# Regulation of Non-Coding RNA in the Growth and Development of Skeletal Muscle in Domestic Chickens

**DOI:** 10.3390/genes13061033

**Published:** 2022-06-09

**Authors:** Hongmei Shi, Yang He, Xuzhen Li, Yanli Du, Jinbo Zhao, Changrong Ge

**Affiliations:** Faculty of Animal Science and Technology, Yunnan Agriculture University, Kunming 650201, China; s18388447069@163.com (H.S.); 18826485012@163.com (Y.H.); manlixuzhen@163.com (X.L.); greedtyas@163.com (Y.D.); zhaojinbo312@163.com (J.Z.)

**Keywords:** domestic chicken, skeletal muscle, non-coding RNA, miRNA, LncRNA, CircRNA

## Abstract

Chicken is the most widely consumed meat product worldwide and is a high-quality source of protein for humans. The skeletal muscle, which accounts for the majority of chicken products and contains the most valuable components, is tightly correlated to meat product yield and quality. In domestic chickens, skeletal muscle growth is regulated by a complex network of molecules that includes some non-coding RNAs (ncRNAs). As a regulator of muscle growth and development, ncRNAs play a significant function in the development of skeletal muscle in domestic chickens. Recent advances in sequencing technology have contributed to the identification and characterization of more ncRNAs (mainly microRNAs (miRNAs), long non-coding RNAs (LncRNAs), and circular RNAs (CircRNAs)) involved in the development of domestic chicken skeletal muscle, where they are widely involved in proliferation, differentiation, fusion, and apoptosis of myoblasts and satellite cells, and the specification of muscle fiber type. In this review, we summarize the ncRNAs involved in the skeletal muscle growth and development of domestic chickens and discuss the potential limitations and challenges. It will provide a theoretical foundation for future comprehensive studies on ncRNA participation in the regulation of skeletal muscle growth and development in domestic chickens.

## 1. Introduction

Chicken is a high-quality animal protein and is the most widely consumed meat product in the world (www.fao.org, accessed on 14 May 2022). Meat is a major end product of poultry production, and the skeletal muscle, which makes up most of the meat and is the most valuable part of chicken meat products, is strongly linked to meat production and quality [1]. Skeletal muscle formation is based on the process of myogenesis, which is a multi-step process [2], and in domestic chicken is separated into two stages: embryonic and postnatal. During the embryonic stage, muscle precursor cells originate from the somite and undergo differentiation and proliferation to form myoblasts, which are induced by specific myogenic transcription factors to form multinucleated myotubes after proliferation, migration, and fusion. Finally, the myotubes functionally mature into fast-twitch and slow-twitch fibers [3,4]. This period is crucial for muscle development, which determines the future number and structure of poultry muscle fibers [5]. After muscle fibers are formed, they undergo hypertrophy at the postnatal stage, which is mostly based on protein conversion (synthesis, degradation, and repair capacity) and the activation of muscle satellite cells (satellite cells proliferate, followed by the fusion of differentiated myoblasts into mature myotubes) [6,7]. In addition to complicated cell developmental processes during muscle fiber formation and hypertrophy, the development of skeletal muscle also depends on the precise regulation of multiple myogenic genes [8].

Although coding-RNAs are primary regulators of cell function, with the development of epigenetics, researchers have found that ncRNAs also play an important regulatory role in skeletal muscle growth and development [4]. ncRNAs are transcribed RNA molecules but do not encode proteins, and they play a role in a wide range of biological processes, including DNA epigenetic modification, transcription, and regulation of gene expression after transcription [9]. This group of ncRNA molecules includes miRNAs, piwi-interacting RNAs, transcription initiation RNAs, small nucleolar RNAs, LncRNAs, and circRNAs [10]. As an important regulator of muscle development, there are abundant ncRNAs in the skeletal muscles of domestic chickens, mainly including miRNA, LncRNA, and CircRNA.

In the last decades, more and more ncRNAs have been found and characterized in domestic chickens (Table 1). Only a few have been studied for their specific mechanisms in skeletal muscle. Consequently, this paper gives an overview of ncRNAs in domestic chicken skeletal muscle to support understanding the molecular genetics of poultry growth and development.

## 2. MiRNA Modulates Skeletal Muscle in Domestic Chicken

MiRNAs are endogenous, non-coding short RNAs that are generally 21–26 nucleotides and evolutionarily conserved [11]. They are extensively distributed in organisms and can regulate genes by causing mRNA translation blockage or degradation via complete or incomplete complementary pairing with target 3′UTRs, as well as binding to the 5′UTR, the CDS region of the target to inhibit gene expression at the post-transcriptional level [12,13]. In contrast, a few investigations have revealed that miRNAs can promote gene expression [14,15]. The miRNAs with spatiotemporal expression patterns form an extraordinarily fine network regulatory mechanism in organisms to play a critical role, and they are dynamic variations in different tissues at distinct periods. A single miRNA can target several mRNAs, and a single mRNA can be regulated by multiple miRNAs [16]. With the widespread use of transcriptomics in livestock production in recent years, a growing amount of research has focused on domestic chicken miRNAs, which range from follicle growth [17], spermatogenesis [18], lipid metabolism [19], muscle development [20], and illness development [21]. Among them, the function of miRNAs in skeletal muscle development includes muscle cell proliferation, differentiation, fusion, apoptosis, and myofiber type specification [22,23]. 

miRNAs are involved in the formation and development of skeletal muscle in domestic chickens and can be classified into the following two types: myoMiRNAs, which are expressed exclusively in muscle tissue, and non-myoMiRNAs, which are expressed in most tissue. Both play a role in skeletal muscle growth and development [24]. Domestic chicken skeletal muscle studies are primarily concerned with the construct of the miRNA library and the function of specific miRNAs across breeds (high vs. low selection broilers [25,26] large vs. small broilers [27], broilers vs. laying hens [28]), developmental stages (embryonic [29], postnatal [30,31], embryonic to postnatal [8]), and skeletal muscle tissues (pectoral muscles, leg muscles). They are involved in a variety of biological processes related to skeletal muscle development and play a crucial role via a complex regulatory network. Among them, functional validation revealed that most miRNAs mainly affect skeletal muscle growth, specifically the proliferation, differentiation, fusion, apoptosis of myoblasts and satellite cells, and the determination of muscle fiber type, as shown in Table 2 and Figure 1.

### 2.1. MiRNA Modulates Skeletal Muscle through Regulating Hormone Levels

Hormones play a crucial role in the development of skeletal muscle, with growth hormone (GH) and insulin-like growth factor-1 (IGF-1) being the most critical hormones [50]. Both factors function in concert with the GH-GHR-IGF1 signaling pathway or independently to promote skeletal muscle growth and increase muscle mass [51,52]. The sex-linked dwarf (SLD) chicken is caused by a recessive mutation of the growth hormone receptor *(GHR)* gene on the Z chromosome [53], which has fewer and smaller muscle fibers than normal size. Once *GHR* gene mutations (most of these mutations are located in the extracellular domain of the *GHR,* where they reduce or abolish the binding affinity to *GH* [54]), they will inhibit myoblast differentiation by inhibiting fusion and promoting migration through the GH-GHR-IGF1 signaling pathway [51]. In 2012, the first miRNA (miRNA let-7b) linked to skeletal muscle development in domestic chickens was discovered, and it was able to suppress skeletal muscle development by targeting the *GHR* gene [32]. In 2016, it was further found that let-7b binds to the 3’UTR region of insulin-like growth factor-2 mRNA binding protein 3 (*IGF2BP3*), reducing IGF2 protein levels in chicken myoblasts. This led to a decrease in chicken myoblast proliferation and cell cycle arrest through the let-7b-IGF2BP3-IGF2 signaling pathway [34]. Similarly, large-bodied Recessive White Rock (WRR) and small-bodied Xinghua chicken (XH) both expressed MiR-146b-3p in their breast muscle tissue. The expression of MiR-146b-3p was significantly higher in small-bodied chickens than in large-bodied ones, which could bind to *GHR* to inhibit the development of skeletal muscle [33].

### 2.2. MiRNA Modulates Skeletal Muscle through Regulating Myoblasts

At the age of 2.5 embryonic days (E2.5), mitotically active myogenic progenitor cells or primary myogenic cells of domestic chicken enter the myotome and express both fibroblast growth factor (FGF) and its receptor, FREK. After E6, they differentiate into satellite cells and myoblasts under the regulation of myogenic differentiation molecules such as myogenic differentiation antigen (*MyoD*), myogenic factor 5 (*Myf5*), and fibroblast growth factor 4 [55]. As previously described, myoblasts, as the precursor cells of muscle fiber, after proliferating and migrating, fuse into myotubes and then differentiate into mature muscle [4]. It is one of the most significant components in embryonic muscle development and endogenous repair, accounting for 2–10% of the total number of myoblast nuclei [56]. Nowadays, many miRNAs have been found to regulate skeletal muscle growth by acting on proliferation [36], fusion [42], differentiation [37], and apoptosis [43,44] of myoblasts in domestic chickens.

#### 2.2.1. Proliferation and Differentiation of Myoblasts

During the embryonic period, myoblasts grow and transform into different types of muscle fibers, which determine the number of muscle fibers after birth [57]. Many studies have shown that miRNAs affect the growth and differentiation of myoblasts in vitro. The expression of MiR-7 and the Krüppel-like factor 4 (*KLF4*) gene were found to be correlated with the development of myoblast in Jinghai yellow chick embryos. Overexpression of MiR-7 will inhibit myoblast proliferation and differentiation by targeting the *KLF4* gene [36]. In the embryonic stage of Haiyang yellow chickens, MiR-214 targets the tRNA methyltransferase 61A (*TRMT61A*) gene to limit proliferation and enhance the differentiation of myoblasts [37]. In the Jianghan chicken, Yin Yang1 (*YY1*) was found to be blocked by MiR-2954, which caused myoblast proliferation and prevented myoblast differentiation into multinucleated myotubes during the embryonic stage [20]. At E10, E12, E14, E16, and E18 in Gushi chickens, MiR-29b-1-5p suppresses myoblast proliferation and promotes its differentiation by targeting ankyrin repeat domain 9 (*ANKRD9*) [38], and the expression of both was negatively correlated at E10-E18. However, Lee et al. [58] demonstrated that *ANKRD9* contains anti-proliferative activity, contradicting this result. This suggests that *ANKRD9* may not be the only target of MiR-29b-1-5p. MiR-30a-3p was found to be differentially expressed at different embryonic stages and was confirmed to target *MYOD*, myocyte generating factor (*MYOG*), and myosin heavy chain (*MYHC*) genes to promote myoblasts differentiation [39]. It was demonstrated that MiR-223 has a dynamic effect on the proliferation and differentiation of myoblasts. MiR-223 could inhibit *IGF2* expression, leading to the inhibition of myoblasts’ proliferation, whereas MiR-223 also suppresses Zinc finger E-box binding homeobox 1 (*ZEB1*) expression under the influence of *MYOD* to promote myoblast differentiation [35]. Some miRNA effects on myoblasts also occur during the postnatal stages. MiR-27b-3p could bind to *MSTN* (1 day old (D1)), 4 weeks old (4w), 8w, and 16w) during the postnatal stages to promote proliferation and inhibit differentiation of myoblasts in the thigh and pectoral muscles [40].

#### 2.2.2. Fusion of Myoblasts

The fusion of myoblasts into myotubes is a critical phase in the formation of skeletal muscle, occurring both during embryonic myogenesis and postnatal muscle regeneration and repair [59]. Fusion, similar to other processes in myogenesis, needs extremely precise spatial and temporal manipulation [60]. In recent years, there has been a lot of attention paid to the fusion process in myogenesis, but very little research about domestic chicken. So far, MiR-140-3p has only been shown to partially inhibit *Myomaker* (a transmembrane protein essential for myoblast fusion) expression in vitro by binding to the 3’ UTR of *Myomaker* to inhibit myoblast fusion in domestic chickens [42].

#### 2.2.3. Apoptosis of Myoblasts

Apoptosis of myoblasts is an important process in myogenesis, and it has been reported that apoptotic myoblasts promote myogenic fusion via the phosphatidylserine receptor (*BAI1*) [61]. However, inhibition of myoblast apoptosis can result in impaired skeletal muscle development, abnormalities, inflammation, and tumorigenesis [45]. Only a few miRNAs have been identified to regulate myoblast apoptosis in domestic chickens, including MiR-16-5p, MiR-146b-3, and MiR-16. During the development of the Xinghua chicken embryo, MiR-16-5p directly targets sestrin 1 (*SESN1*) to regulate the p53 signaling pathway, enhance myoblast differentiation, and inhibit apoptosis [43]. The MiR-146b-3 inhibits *MDFIC* (Negatively regulator of MyoD family transcription during fibroblast differentiation) and the PI3K/AKT pathways in the XH chick embryo to promote myoblast apoptosis [45]. The MiR-16 expression was significantly reduced in hypertrophied chicken breast muscles compared to normal ones and inhibits myoblasts’ proliferation and promotes apoptosis by directly targeting B cell lymphoma-2 (*Bcl2*) and Forkhead box transcription factor O1 (*FOXO1*) [44].

### 2.3. MiRNA Modulates Skeletal Muscle through Regulating Satellite Cells

Skeletal muscle satellite cells (SMSCs) are stem cells with proliferating and differentiating abilities that live in adult skeletal muscle and can repair or regenerate damaged muscle [62]. Satellite cells are relatively quiescent under normal conditions due to the expression of the paired box transcription factor (*Pax7*). Once suffering from stress, such as heavy loads, trauma, and so on, the expression of *Pax7* and *MyoD* would change, causing the cells to enter the proliferative phase to produce numbers of myoblasts. Myoblasts are further differentiated and fused to form myotubes by increased expression of myogenic regulatory factors (MRFs) to contribute to skeletal muscle repair and regeneration [63,64]. MiRNAs are critical for skeletal muscle regeneration not only because they help keep SMSCs dormant, but they regulate their proliferation, differentiation, and apoptosis.

#### 2.3.1. Proliferation and Differentiation of Satellite Cells

In domestic chicken, MiR-3525, MiR-99a-5p, MiR-9-5p, and MiR-21-5p are currently identified miRNAs that act on SMSCs’ proliferation and differentiation as follows: SMSCs proliferation and differentiation are inhibited by MiR-3525, which targets the PDZ and LIM domain 3 (*PDLIM3*) and the p38/MAPK signaling pathway [46]. MiR-99a-5p, which targets myotubularin-related protein 3 (*MTMR3*), stimulates the proliferation of SMSCs while inhibiting differentiation [47]. MiR-9-5p inhibits SMSC proliferation and differentiation by targeting IGF2BP3 via *IGF-2* and activating the PI3K/Akt signaling pathway [27]. MiR-21-5p stimulates the proliferation and differentiation of SMSCs by targeting Krüppel-like factor 3 (*KLF3*) [48].

#### 2.3.2. Apoptosis of Satellite Cells

MiRNAs that act on SMSC apoptosis include MiR-200a-3p and MiR-148a-3p. The expression of MiR-200a-3p was found to be upregulated at embryo stages E10, E13, E16, and E19 in broiler and laying hen pectoral muscles and was significantly higher in broilers than in laying hens. The functional validation in vitro demonstrated that it could target transforming growth factor 2 (*TGF-2*) to regulate the TGF-2/SMAD signaling pathway, accelerating the differentiation and proliferation of chicken SMSCs and inhibiting apoptosis [28]. MiR-148a-3p downregulates the expression of mesenchymal homology frame 2 (*Meox2*) and activates the PI3K/AKT signaling pathway to promote SMSC differentiation and inhibit apoptosis without affecting proliferation [49].

### 2.4. MiRNA Modulates Skeletal Muscle through Specificating and Maintaining Muscle Fiber Type

Muscle fibers in domestic chicken can be characterized by metabolism-based as oxidative (types I and IIA) and glycolytic (type IIB) or as slow (type I) and fast (types IIA and IIB) based on contraction rate [65]. Different muscle fiber types affect meat quality characteristics such as color, tenderness, water-holding capacity, juiciness, and flavor [66]. Although the physiological and functional differences between muscle fiber types have been widely explored, the molecular regulation of the various muscle fiber types in chickens remains mostly unclear, with even fewer instances involving ncRNAs. MiRNA modulation of diverse muscle fiber types in chickens is still in its infancy. Liu et al. (2020) systematically compared the mRNA and miRNA transcriptomes of oxidative muscle sartorius (SART) and glycolytic muscle pectoralis major (PMM) in Chinese Qingyuan partridge chickens using RNA sequencing. They found that MiR-499-5p and MiR-196-5p were most abundant and upregulated in SART and demonstrated that MiR-499-5p targets *SOX6* (a repressor of slow muscle-specific gene expression), whereas MiR-196-5p targets *CALM1* (a key component of the cGMP-PKG and calcium signaling pathways), both together regulating slow muscle fiber formation [1].

## 3. LncRNA Modulates Skeletal Muscle in Domestic Chicken

LncRNAs are a new type of regulatory RNA that is longer than 200 bp and is transcribed by RNA polymerase II. They account for approximately 87 percent of all ncRNAs and play an important role in muscle development by regulating transcriptional and post-transcriptional [67,68]. LncRNAs are important members of the regulatory network of skeletal muscle development and have been demonstrated to affect skeletal muscle proliferation and differentiation via competing for endogenous RNAs (ceRNAs), such as binding to miRNA and inhibiting its functions. Lnc-MD1 is the first LncRNA that has been linked to myogenesis and regulates the expression of Mastermind-like 1 (*MAML1*) and myocyte enhancer factor2C (*MEF2C*) by targeting MiR-133 and MiR135, which play an important part in the temporal control of human myogenic cell development [69]. In addition, some LncRNAs regulate gene expression in cis or trans, such as Lnc-EDCH1. It could improve mitochondrial efficiency by activating the AMPK pathway via *SERCA2*, which regulates myoblast proliferation and differentiation in vitro, reduces intramuscular fat deposition, activates the slow muscle phenotype, and inhibits muscle atrophy in vivo [70]. Furthermore, several LncRNAs can influence muscle development and atrophy by modifying proteins. Jin et al. (2018) found that Lnc*-SYISL* (*SYNPO2* intron sense overlapping LncRNA) could interact directly with the enhancer of the PRC2 (core component zeste homolog 2 protein) to regulate the expression of p21 and muscle-specific genes, promoting myoblast proliferation and inhibiting myogenic differentiation [71]. Although LncRNAs are associated with skeletal muscle development in livestock (pigs [72], sheep [73,74], goats [75], donkeys [76], buffalos [77], and domestic chickens [78]) have been identified, fewer mechanisms of them have been explored.

Only 6 LncRNAs with myogenic functions have been identified currently in the skeletal muscle development of domestic chickens, including proliferation, differentiation, and fusion of skeletal muscle stem cells, as well as muscle hypertrophy and fiber type conversion [79]. Li et al. (2012) [80] were the first to successfully identify LncRNAs in chicken pectoral muscle during embryonic development using RNA-seq. Although they identified 281 new intergenic LncRNAs in the chicken genome, they did not predict LncRNA target and function validation. Li et al. (2016) [81] identified a total of 129, 132, and 45 differentially expressed LncRNAs (E11 vs. E16, E11 vs. D1, and E16 vs. D1) in XH chicken leg muscle. Moreover, they identified the cis-and trans-regulatory targets of differentially expressed LncRNAs and constructed the lncRNA-gene interaction networks. Furthermore, they analyzed the LncRNA transcriptome of Shouguang chickens from embryonic stage (E12, E17) to post-hatching (D1, D14) and identified the following two profiles with opposite expression trends: profile 4 with a down-regulation pattern and profile 21 with an up-regulation pattern. According to functional analysis of the targets, profile 4 contributes to cell proliferation, while profile 21 is primarily involved in metabolism [67]. Li et al. (2021) [82] established 12 RNA libraries in postnatal Gushi chicken pectoral muscle (6 w, 14 w, 22 w, and 30 w), and they found dynamic changes in LncRNA expression in pectoral muscle at various stages, suggesting that some LncRNAs that target *MEF2C* may be involved in muscle regulation via the MAPK signaling pathway. Without a doubt, with the development of sequencing technology and bioinformatics tools, the research related to LncRNA in domestic chickens will become mature. Although a considerable number of LncRNAs associated with skeletal muscle development in domestic chickens have been identified, only a small number of LncRNAs have been demonstrated to function at the molecular cellular level. Based on the existing studies, the role of LncRNA in domestic chicken skeletal muscles is included as a cis or trans regulator, or by sponging competing miRNAs and encoding short molecular micropeptides to regulate gene expression. Hence, we summarized the information about the expression and functions of LncRNAs in the skeletal muscles of domestic chickens, as shown in Table 3 and Figure 2.

### 3.1. LncRNA Modulates Skeletal Muscle through Sponging miRNAs on Myoblasts 

Six homology frame 1 (*Six1*) is highly expressed in slow muscle fibers, where it promotes the conversion of fast muscle fibers to slow muscle fibers and was discovered to contain a potential binding site for LncRNA-Six1 and MiR-1611. Thus, LncRNA-Six1 could regulate *Six1* expression and fiber type switching by competing with MiR-1611 [68]. The Lnc-IMFNCR could act as a ceRNA by competing with MiR-128-3p and MiR-27b-3p, upregulating peroxisome proliferator-activated receptor gamma (*PPARG*), and contributing to the development of intramuscular adipocytes in Gushi chicken [83]. Lnc-IRS1 controls myoblast proliferation and differentiation in vitro as well as muscle mass and muscle fiber numbers in vivo, and its expression upregulates with myogenic differentiation. As a ceRNA for MiR-15a, MiR-15b-5p, and MiR-15c-5p, Lnc-IRS1 regulates the expression of the *IGF1* downstream receptor, insulin receptor substrate 1 (*IRS1*). It is upregulated in hypertrophic broiler chickens and promotes myoblast proliferation and differentiation, as well as activating the IGF1-PI3K/AKT pathway to prevent muscle atrophy [84].

### 3.2. LncRNA Modulates Skeletal Muscle through Regulating Gene Expression in Cis or Trans

The LncRNA-Six1, which is situated 432 bp upstream of the *Six1* gene encoding region, was differentially expressed between WRR and XH chickens. Overexpression of LncRNA-Six1 will enhance the expression of muscle growth-related genes (*MYOG, MYHC, MYOD, IGF1R*, and *INSR*), and it encodes a micropeptide that affects *Six1* protein expression in a cis-regulatory manner, promoting myoblast proliferation [78]. In addition, LncRNA-FKBP1C, which is differentially expressed between WRR and XH chickens, can bind to *MYH1B* and enhance its protein stability by cis-regulation. It can also inhibit myoblast proliferation or apoptosis and promote differentiation as well as reduce the expression of fast muscle genes and increase slowly [85].

In addition, Lnc-EDCH1 is abundantly expressed in skeletal muscle cells of WRR. It could increase calcium transport ATPase (SERCA2) protein stability to promote myoblast proliferation and inhibit differentiation, induce a slow muscle phenotype and inhibit muscle atrophy, regulate Ca2+ homeostasis, and activate the AMPK pathway to improve mitochondrial efficiency in skeletal muscle cells [70].

## 4. CircRNA Modulates Skeletal Muscle in Domestic Chickens

CircRNA is a non-coding RNA that is abundant in tissues (e.g., in the human and mouse brains) and cells and has a covalent closed-loop structure [86]. CircRNAs may be made from any part of the genome and hence have a wide range of lengths. They also lack free 5’ and 3’ ends, making them extremely conserved and stable [87], as well as resistant to RNA exonucleases (RNase) [88,89]. CircRNAs influence gene expression at multiple levels, including transcription (via RNA binding proteins (RBPs) and miRNAs), pre-mRNA splicing, translation, and self-translation into proteins [90,91]. In addition, circRNAs have been associated with biological processes such as cell proliferation, survival, and differentiation [92]. Although various circRNAs were found between the 1970s and 1990s, it was not until the advent of high-throughput RNA sequencing (RNA Seq) that their quantity and function attracted attention, and worldwide investigation of circRNAs began [93]. Hansen et al. (2013) conducted the first functional investigation of naturally occurring CircRNA (CiRS-7) and discovered substantial suppression of MiR-7 activity, which increased MiR-7 target expression [94]. Numerous studies have shown that circRNAs play an important role in skeletal muscle control, mainly sponging miRNAs and moderating the inhibitory effects of miRNAs on mRNAs [95]. For example, Circ-CDR1 promotes satellite cell development by sponging MiR-7, which inhibits *IGF1R* expression [96]. Circ-HIPK3 is involved in promoting C2C12 myoblast proliferation and differentiation through the MiR-7 and transcription factor 12 (*TCF12*) axis [97]. Circ-ARID1A promotes skeletal muscle regeneration by targeting MiR-6368 [98] and affects skeletal muscle strength [99]. 

CircRNAs are found in a wide range of organs and cell types in domestic chickens, and they play a role in follicular development [100,101], bursal development [102], ventral lipid deposition [103], and skeletal muscle development [104]. Chicken circRNAs have shorter transcripts and similar GC content to mRNAs and LncRNAs [105], and CircRNA levels are generally lower than the corresponding host genes [106]. However, some circRNAs are expressed significantly higher than linear transcripts in some unique cell lines or tissues [107]. Numerous studies have demonstrated that circRNAs are abundant in skeletal muscle and are involved in myogenesis. Thus, we summarize the known functions and molecular mechanisms of circRNAs in domestic chicken skeletal muscle in Table 4 and Figure 2. Among them, CircRNA regulates domestic chicken skeletal muscle development by the following: (1) regulating its host gene expression to regulate skeletal muscle development; (2) sponging miRNA to mitigate its inhibitory effect on mRNA; (3) translating into protein to directly regulate skeletal muscle development.

### 4.1. CircRNA Modulates Skeletal Muscle through Regulating Parental Genes

Circ-GHR is abundant in the nucleus of myoblasts and is derived from the chicken *GHR* gene. Circ-GHR is reduced in the leg and pectoral muscles steadily from E13 to 7 w. It is positively correlated with GHR, and overexpressing Circ-GHR will promote myoblast proliferation. It is hypothesized that Circ-GHR may promote myoblast proliferation by regulating the expression of *GHR* mRNA and *GH* binding protein (*GHBP*), but it has been demonstrated to have no significant effect in DF-1 cell lines, so the definitive mechanism needs to be further explored [108].

### 4.2. CircRNA Modulates Skeletal Muscle through Sponging miRNAs on Myoblasts

There was a significant difference in the expression of Circ-SVIL in the skeletal muscles of E11, E16, and D1 Xinghua chickens. It was upregulated dramatically between E11 and E14 and stayed at a high level into late embryonic development. Further functional validation demonstrated that it promoted chicken myoblast proliferation and differentiation by sponging MiR-203 (MiR-203 was differentially expressed during chicken embryonic skeletal muscle development, being particularly abundant in E12 and E14 [118]) and upregulating the mRNA levels of transcription factors c-JUN and Myocyte enhancer factor 2C (*MEF2C*) [109]. In 7 w XH chickens, a total of 532 circRNAs were differently expressed between the pectoralis major (PEM) and the soleus (SOL). By sponging MiR-499-3p, Circ-PTPN4 regulated *NAMPT* expression, which activated AMPK signaling and led to more myoblast growth and differentiation while suppressing mitochondrial biogenesis and activating the fast muscle fiber phenotype [110]. Circ-RBFOX2s have been demonstrated to promote myoblast proliferation by binding MiR-206 in the embryonic leg muscles of XH chickens [111]. Circ-HIPK3 expression was differentially expressed in the skeletal muscle of E11, E16, and D1 Yuhe chickens and promoted myoblast proliferation and differentiation by eliminating MiR-30a-3p binding to *MEF2C* [112]. Circ-ITSN2, a chicken intersectin 2 (*ITSN2*)-derived gene, was expressed at a higher level in the fast muscle-growth broiler (ROSS 308 broiler) than in the slow muscle-growth laying hen (White Longhorn layer) and was sustained at a high expression. It was further revealed that Circ-ITSN2 boosts chicken myoblast proliferation and differentiation by alleviating the MiR-218-5p targeting domain protein LIM domain only 7 (*LMO7*) [113].

### 4.3. CircRNA Modulates Skeletal Muscle through Sponging miRNAs on Satellite Cells

The pectoral muscles of 12 broilers and 12 laying hens were sequenced at four different embryonic stages (E10, E13, E16, and E19), and 228 circRNAs were identified that were differentially expressed between broilers and laying hens, with 43 circRNAs that were significantly differentially expressed across multiple embryonic stages [114]. Further, it was found that Circ-TMTC1 inhibits SMSC differentiation via adsorption of MiR-128-3p, which inhibits C2C12 myoblasts’ growth by targeting muscle growth inhibitor mRNA but helps with myotube formation. In fast-growing broiler chickens, Circ-PPP1R13B expression is significant, and it promotes SMSC proliferation and differentiation by upregulating the expression of the MiR-9-5p target gene *IGF2BP3* and activating the downstream IGF/PI3K/AKT signaling pathway [115]. By binding to MiR-204, Circ-FNDC3AL enhances the proliferation and differentiation of chicken SMSCs by upregulating lymphoma 9 (*BCL9*) expression [116].

### 4.4. CircRNA Modulates Skeletal Muscle through Translating Directly into Protein

CircRNAs are categorized as non-coding RNAs because they lack 5’ and 3’ ends and are ineligible for translation [119]. Nevertheless, most circRNAs from exons are found in the cytoplasm, implying that circRNA translation is possible [120]. As the research advanced, it was discovered that circRNAs may be translated into functional peptides in the presence of internal ribosome entry sites (IRES) and open reading frames (ORF). CircRNAs, on the other hand, have a lower potential translation capacity than linear mRNAs [91]. Pamudurti et al. (2017) [90], provided the first direct proof that Circ-Mbl, a product of the *Mbl* locus, could be translated into endogenous protein in Drosophila. In mouse and human myoblasts, Circ-ZNF609 was found to be linked with heavy polysomes and converted to proteins in a splicing-dependent and cap-independent way [91]. Following that, various translatable circRNAs, such as Circ-FBXW7, Circ-PPP1R12A, Circ-SHPRH, and Circ-AKT3, were discovered and play essential roles in cancer cell development [121,122,123].

There were occurrences of CircRNA being directly translated into protein in domestic chickens, but the only data available for the modification of skeletal muscle development is Circ-FAM188B. Circ-FAM188B is a stable cyclic RNA that is variably expressed in broiler and laying hen embryos during skeletal muscle development and has demonstrated a distinct pattern of a substantial decline in expression from E10 to D35. It is predicted by the bioinformatics tools that Circ-FAM188B contains an ORF and has the coding potential to encode Circ-FAM188B-103aa, which has also been confirmed by the existence of the IRES. Experiments showed that the function of Circ-FAM188B-103aa in chicken myogenesis was the same as that of its host gene transcript, FAM188B, which enhanced proliferation but restricted differentiation of chicken SMSCs [117].

Currently, studies on the mechanism of CircRNA regulation of muscle development have focused on the interaction between CircRNA and miRNA. Due to the extensive study of circRNAs, it has been demonstrated that SNPs can influence the generation of circRNAs and their expression levels. SNPs linked to multiple cases of sclerosis, such as those found in the *STAT3* gene, influence the expression level of Circ-0043813 [124]. The Circ-FOXO3 flanking intron rs12196996 polymorphism affects Circ-FOXO3 expression and raises the risk of coronary artery disease [125]. According to related studies, SNPs have an effect on the degree of expression of circRNAs in domestic chickens. Circ-TAF8, a ubiquitous and differently expressed CircRNA in chicken embryonic leg muscles, is the product of head-to-tail cyclization of exons 2, 3, 4, and 5 of the protein-coding gene *TAF8* on chromosome 26, which promotes proliferation and inhibits differentiation of chicken myoblasts. The SNPs in the introns on both sides of the Circ-TAF8 gene were also analyzed for their connection with chicken carcass features in 335 partridge chickens. Eight SNPs were related to carcass traits (leg muscle weight, live weight, half-bore weight, and full-bore weight), all of which had short complementary sequences, implying that polymorphisms at these SNP loci may affect the Circ-TAF8 production [107].

## 5. Prospect

The broiler breeding business has long sought to improve meat yield and quality. For both of those, an understanding of the developmental condition of skeletal muscle and related regulatory processes is required. However, research on ncRNA is still in the early stages in the skeletal muscles of domestic chickens currently.

The process of miRNA analysis can be basically summarized as the following steps: sequencing, target prediction and validation, expression pattern, pathway network, and functional validation [126]. Those are the basis of existing miRNA databases for data inclusion and functional annotation. However, there were some problems that limited the further use of these databases in the process of chicken miRNA analysis. To begin with, there are numerous database types that fail to effectively integrate. Most databases only offer partial information about miRNA, which will reduce the efficiency of the miRNA data analysis. Although some databases integrate multiple functions, such as mirTools 2.0 or CPSS, they are data-lagging and poorly personalized. Second, some databases are no longer available because of a lack of long-term maintenance and updates. The future development of miRNA databases should be an integration of functions, highly autonomous, continued data updates, and stable technological support.

Compared to miRNAs, chicken LncRNA research is still at an earlier stage. On the one hand, this is due to the deficiencies of the strategies for LncRNA identification and functional annotation as follows: (1) The low conservativeness of LncRNA sequences among chicken species leads to the possibility of recognition barriers due to indels in the sequence, even if it is in the same position in the genome [127]. (2) The existing LncRNA databases are mainly focused on humans and mice, but there is no specific database for chickens with tissue-specific expression patterns of LncRNA. This could lead to some LncRNAs with specific spatiotemporal expression patterns or unannotated LncRNAs being ignored. On the other hand, the complexity of LncRNAs in organisms leads to limited and costly validation, as follows: (1) A loss of function experiment is an effective method to verify LncRNA function and include RNA interference (RNAi), antisense oligonucleotides (ASOs), and genomic manipulation techniques. RNAi and ASOs both inhibit RNA function by binding to RNA and causing RNA degradation, but their limited targets and low silencing efficiency affect their application in LncRNA [128]. (2) The genomic manipulation techniques include Zinc Finger Nucleases (ZFNs), Transcription Activator-Like Effector Nucleases (TALENs), and the Clustered Regularly Interspersed Short Palindromic Repeat (CRISPR) system. In particular, the CRISPR system has become a revolutionary tool in molecular biology due to its usability and flexibility [129]. However, the limited editing strategy and poor editing efficiency limit the application of the CRISPR/Cas9 editing system in chickens [130]. (3) LncRNAs can influence phenotypes directly or indirectly through different pathways. Therefore, for the functional validation of LncRNA, a comprehensive verification of its possible biological functions is required. For example, LncRNABMP4 can promote the expression of *BMP4*, a developmental and migration-related gene in PGCs (precursor gonadal germ cells), by repressing the expression of MiR-12211 and can also directly promote the transcription of *BMP4* by encoding the small peptide EPC5 [131]; Similarly, in PGCs, LncPGCAT-1 (LncRNA PGC transcript-1), which directly regulates the expression of the chicken vasa homologue (*Cvh*) gene and *C-Kit* to promote the formation of PGCs. It also promotes the expression of *MAPK1* (mitogen-activated protein kinase (1) by inhibiting the binding of MiR-1591 to *MAPK1*, which ultimately promotes the development of PGCs. These studies suggest that LncRNAs may influence the occurrence of the same phenotype through multiple pathways [132]. Therefore, during the functional validation of LncRNA, the results obtained from a single validation target are often insufficient to explain the full function of LncRNA, while the simultaneous validation of multiple functions causes an increase in time and costs.

Currently, the methods used to identify the presence of circRNAs mainly include PCR, northern blot, etc. However, it is still a challenge to annotate the function of circRNAs [133]. The commonly used loss-of-function experiments are more limited in the process of verifying the function of circRNAs. This is because functional silencing of circRNAs often interferes with their linear RNA expression, ultimately leading to a misunderstanding of the results [95]. Therefore, until the impact of knockdown of circRNAs on their host genes is effectively addressed, the strategy for multi-database predicted CircRNA function will remain the mainstream for a long time [134,135].

Despite a large number of new ncRNAs having been identified and discovered, only a small number of ncRNAs have been studied clearly for their functions and mechanisms. Therefore, the solution to the problem of functional annotation and functional validation will be the focus and difficulty of future research in domestic chicken skeletal muscle ncRNA research.

## Figures and Tables

**Figure 1 genes-13-01033-f001:**
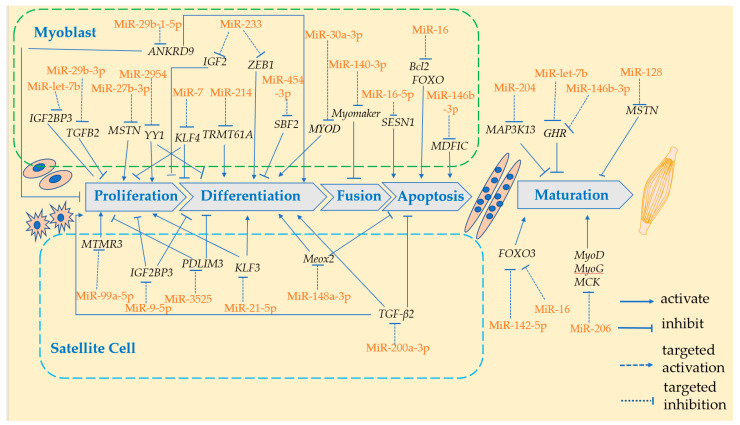
The role of miRNA in the growth and development of skeletal muscle in domestic chicken. MiRNA, shown in orange in the schematics, promotes or inhibits the developmental process of skeletal muscle by targeting mRNAs (shown in black italics) to cause their translation to be blocked or degraded. The majority of those miRNAs were involved in the regulation of myoblast proliferation, differentiation, fusion, and apoptosis, as well as satellite cell proliferation, differentiation, and apoptosis. Otherwise, minority miRNAs regulate muscle fiber maturation—the hypertrophy and mass changes of muscles—by targeting hormone-related genes and myogenic factors in skeletal muscle.

**Figure 2 genes-13-01033-f002:**
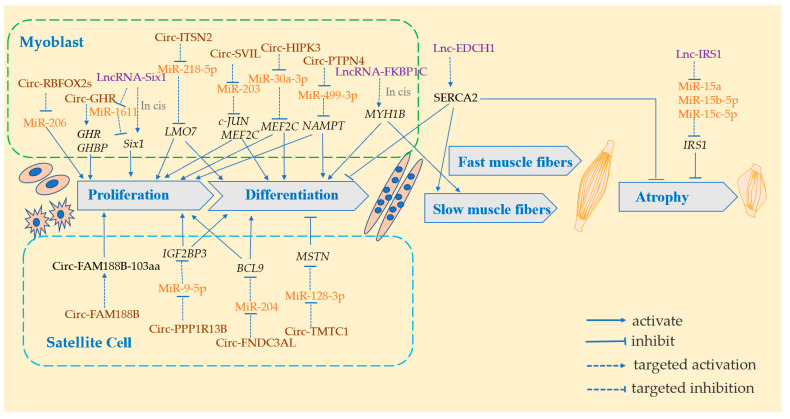
The role of LncRNA and CircRNA in the growth and development of skeletal muscle in domestic chickens. LncRNAs are shown in purple and they promote or suppress various stages of skeletal muscle development (including myoblast proliferation, differentiation, slow muscle fiber formation, and muscle atrophy) by sponging miRNA (miRNA shown in orange) or by cis-regulating the expression of genes (genes in black italics) to promote or inhibit all stages of skeletal muscle development (including myoblast proliferation, differentiation, slow muscle fiber formation, and muscle atrophy) in domestic chicken by alleviating the repression of target genes. CircRNA is shown in brown in the figure and functions to promote or inhibit the proliferation and differentiation of myoblasts and satellite cells. Their functions are through the following three ways: direct regulation of parental genes; sponging miRNA (miRNA shown in orange) to alleviate its repression of target genes; direct translation of CircRNA itself into proteins. (genes shown in black italic, proteins shown in black non-italic).

**Table 1 genes-13-01033-t001:** Numbers of ncRNAs in domestic chicken.

Items	miRNA	lncRNA	circRNA
Chicken	674	12850	494
Reference Database	miRBase (https://www.mirbase.org/, accessed on 14 May 2022)	NONCODE (http://www.noncode.org/, accessed on 14 May 2022)	CircFunBase (http://bis.zju.edu.cn/CircFunBase/index.php, accessed on 14 May 2022)

**Table 2 genes-13-01033-t002:** Species and functions of miRNAs involved in skeletal muscle of domestic chicken.

Regulating Method	miRNA	Target Genes	Function	Chicken Breeds	Organs	Days of Age	References
Hormone-related genes	MiR-let7b	*GHR*	Inhibit the growth of skeletal muscle	Recessive White Rock and Xinghua	Leg muscles	E14, 7 w	[32]
	MiR-146b-3p				Pectoral muscles	7 w	[33]
Myogenic factors	MiR-204	*MAP3K13*	Inhibit skeletal muscle growth	Sichuan mountainous black-bone and Dahen	Pectoral muscles	10 w	[25]
	MiR-142-5p	*FOXO3*	Promotes skeletal muscle growth	Recessive White Rock and Xinghua	Pectoral and leg muscles	7 w	[26]
	MiR-128	*MSTN*	Muscle mass loss	Sex-linked dwarf	Leg muscles	E14, 7 w	[34]
	MiR-206	*MyoD* *MyoG* *MCK*	Increase muscle mass	F2(Recessive White Rock and Xinghua)	Pectoral and leg muscles	30 w	[35]
Proliferation and differentiation of myoblasts	MiR- 29b-3p	*TGFB2*	Inhibits proliferation	Shouguang	Pectoral muscles	E12, E17, D1, D14, D56, D98	[8]
	MiR-2954	*YY1*	Promotes proliferation and inhibits differentiation	Jinghai	Leg muscles	E7, E10, E13, E15, E18, D1	[20]
	MiR-7	*KLF4*	Inhibition proliferation and differentiation	Jinghai	Pectoral and leg muscles	E12, E14, E16, E18, E20, D1	[36]
	MiR-214	*TRMT61A*	Promotes differentiation	Haiyang	Pectoral and leg muscles	E12, E14, E16, E18, D1	[37]
	MiR-29b-1-5p	*ANKRD9*	Inhibits proliferation and promotes differentiatio	Gushi	Pectoral muscles	E10, E12, E14, E16, E18	[38]
	MiR-30a-3p	*MYOD*	Promotes differentiation	Gushi	Pectoral muscles	E10, E12, E14, E16, E18	[39]
	MiR-233	*IGF2* *ZEB1*	Inhibits proliferation and promotes differentiation	Recessive White Rock and Xinghua	Leg muscles		[35]
	MiR- 27b-3p	*MSTN*	Promote proliferation	Hangyang	Pectoral and leg muscles	E12, E16, E18, E20, D1, 4W, 8W, 16W	[40]
	MiR-454-3p	*SBF2*	Inhibits differentiation	Tibetan	Pectoral muscles	D300	[41]
Fusion of myoblasts	MiR-140-3p	*Myomaker*	Inhibit fusion	Chick	Leg muscles	E10	[42]
Apoptosis of myoblasts	MiR- 16-5p	*SESN1*	Inhibits proliferation and differentiation, promotes apoptosis	Xinghua	Pectoral and leg muscles	E10- E20, D1	[43]
	MiR-16	*Bcl2 FOXO1*	Inhibits proliferation and promotes apoptosis	Commercial and heritage chickens	hypertrophic and normal pectoral muscle	7 w	[44]
	MiR-146b-3p	*MDFIC*	Inhibits proliferation and differentiation, promotes apoptosis	Xinghua	Leg muscles	E11, E16	[45]
Proliferation and differentiation of satellite cells	MiR-9-5p	*IGF2BP3*	Inhibits proliferation and differentiation	Recessive White Rock and Xinghua	Pectoral muscles	7 w	[27]
	MiR-3525	*PDLIM3*	Inhibits proliferation and differentiation	Ross 308	Pectoral muscles	D4	[46]
	MiR-99a-5p	*MTMR3*	Promote proliferation	Ross 308	Pectoral muscles	D4	[47]
	MiR-21-5p	*KLF3*	Promotes proliferation and differentiation	Dahen	Leg muscles	D3	[48]
Apoptosis of satellite cells	MiR-499-5p/MiR-196-5p	*SOX6/CALM1*	Regulation of slow muscle fiber formation	Qinguan	PMM and SART	D140	[1]
	MiR-200a-3p	*TGF-β2*	Promotes proliferation and differentiation, inhibits apoptosis	Ross 308 and White Longhorn	Pectoral muscles	E10, E13, E16, E19,	[28]
	MiR-148a-3p	*Meox2*	Pomotes differentiation and inhibits apoptosis	Ross 308	Pectoral muscles	D4	[49]

**Table 3 genes-13-01033-t003:** Species and functions of LncRNAs involved in skeletal muscle of domestic chicken.

Regulating Method	Lnc RNA	Ce RNA	Target Genes	Functions	Chicken Breeds	Organs	Days of Age	References
Sponges miRNAs	Lnc-Six1	MiR-1611	*Six1*	Promotes myoblast proliferation and division	Xinghua	Pectoral and leg muscles	7 w	[68]
	Lnc-IMFNCR	MiR- 128-3p/ MiR- 27b-3p	*PPARG*	Promotes intramuscular adipocytes differentiation	Gushi	Pectoral and leg muscles	3 w	[83]
	Lnc-IRS1	MiR-15a/miR-15b-5p/MiR-15c-5p	*IRS1*	Inhibits muscle atrophy	Hypertrophic broilers and leaner broilers	Pectoral muscles	6 w	[84]
Regulation of gene expression in cis or trans	Lnc-FKBP1C		*MYH1B*	Inhibits myoblast proliferation and promotes differentiation, upregulates slow muscle genes	Recessive White Rock and Xinghua	Pectoral and leg muscles	7 w	[5]
	Lnc-EDCH1		*SERCA2*	Promotes myoblast proliferation, Inhibits differentiation, activates slow muscle phenotype, reduce muscle atrophy	Recessive White Rock and Xinghua	Leg muscles	D1, D4, D8	[70]
	Lnc-Six1		*Six1*	Promotes myoblast proliferation and division	Recessive White Rock and Xinghua	Pectoral and leg muscles	7 w	[78]

**Table 4 genes-13-01033-t004:** Species and functions of circRNAs involved in skeletal muscle of domestic chicken.

Regulating Method	CircRNA	Target miRNAs	Target Genes	Functions	Chicken Breeds	Organs	Days of Age	References
Regulating parental genes	Circ-GHR		*GHR* *GHBP*	Promotes myoblast proliferation	Xinghua	Pectoral and leg muscles	E13, E16, E19, D1, 1 W, 2 W, 3 W, 4 W, 5 W, 6 W, 7 W	[108]
Sponging miRNAs act on myoblasts	Circ-SVIL	MiR-203	*c-JUN MEF2C*	Promotes myoblast proliferation and differentiation	Xinghua	Leg muscles	E10-D1	[109]
	Circ-PTPN4	MiR-499-3p	*NAMPT*	Promotes myoblast proliferation and differentiation	Xinghua	Pectoralis major and soleus	7 w	[110]
	Circ-RBFOX2s	MiR-206		Promotes myoblast proliferation	Xinghua	Leg muscles	E10-D1	[111]
	Circ-HIPK3	MiR-30a-3p	*MEF2C*	Promotes myoblast proliferation and differentiation	Yuhe	Leg muscles	E10-D1	[112]
	Circ-ITSN2	MiR-218-5p	*LMO7*	Promotes myoblast proliferation and differentiation	Ross 308 and White Longhorn	Pectoral and leg muscles	E10, E13, E16, E19	[113]
Sponging miRNAs act on satellite cells	CircTMTC1	MiR-128-3p	*MSTN*	Inhibition of satellite cell differentiation	Ross 308 and White Longhorn	Pectoral muscles	E10, E13, E16,E19	[114]
	Circ-PPP1R13B	MiR-9-5p	*IGF2BP3*	Promotes satellite cell proliferation and differentiation	Ross 308 and White Longhorn	Pectoral muscles		[115]
	Circ-FNDC3AL	MiR-204	*BCL9*	Promotes satellite cell proliferation and differentiation	Ross 308	Pectoral muscles	E10, E13, E16, E19	[116]
Translating directly into protein	Circ-FAM188B		CircFAM188B-103aa	Promotes satellite cell proliferation and inhibits differentiation	Ross 308	Pectoral and leg muscles	D1, D3, D5, D7, D14, D21, D28, D35	[117]

## Data Availability

Not applicable.

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
