# Peer review of "Regulation of Non-Coding RNA in the Growth and Development of Skeletal Muscle in Domestic Chickens"

_genes, 2022, doi:10.3390/genes13061033_

Round 1
Reviewer 1 Report
In the manuscript, Genes-1717898 by Shi et al, authors have reviewed literature about the non-coding RNAs including miRNA, lncRNAs and circRNAs in the growth and development of skeleton muscles in domestic chickens. Given the widespread use of domestic chicken as a livestock and poultry production, this review would help understanding the current state of knowledge along with the scope and directions for the future studies. Authors have made efforts to cover most of the relevant studies and cited them appropriately. Providing tables and schematics helped to quickly grab the information. In general, this review article is comprehensive and informative, however there are certain points that would need to be addressed-
Major-
- The current title “The regulation network of non-coding RNA in domestic chicken skeletal muscle” needs to be revised. “regulatory network” in place of “regulation network”.
- “in domestic chicken skeletal muscle” can be revised with “in the growth and development of skeletal muscle in domestic chicken”.
- Most of the heading titles are not informative. For example, line 84, subheading 2.1 title is “Hormone levels” and doesn’t not hint at all what is going to be covered in this subheading. Moreover, the preceding paragraph does not clearly put the foundation of subheadings. Similarly, line 103, subheading “Myoblasts” and most of the subheading onwards are vague. Please consider rephrasing them to make meaningful titles.
- Schematic diagrams are not well defined. It would be helpful in authors provide a brief description of the schematic diagrams in the corresponding legends.
Minor-
- In line 20. “discuss the limitations and challenges of the current related studies.”. What do authors imply with “current related studies”? Please revise accordingly.
- The very first paragraph conveys many messages but cites on one study at the end. Please provide references for the support of literature used.
- Line 48-49. “In the last few years, more and more ncRNAs have been found and characterized in 48 chicken skeletal muscle.” Please provide reference.
- Heading numbering needs to be revised for accuracy. After Heading 2 and its subheading, numbering for headings and subheading is wrong. For example, Line 311, heading number should be 3 in place of 2.
- Numbering for tables is also incorrect. Please revise for the correctness.
- The column if “Year” in the tables is not necessary as the referenced study is cited there. Please remove it from all the tables.
- Line 229. Please define the ce RNA.
- Please consider replacing MiRNA with miRNAs throughout the manuscript. Similarly, lncRNAs can be replaced with LncRNAs for consistency.
Reviewer 2 Report
Shi et al. reviewed the roles of ncRNAs in the domestic chicken skeletal muscle. The authors have collected enough materials or papers for building up the manuscript. However, the manuscript still needs to be improved.
- The prospect is vague and misleading- it shows the authors did not understand enough about the field. Some statements are too generalized and they might cause more confusion for the readers.
- The title is misleading as well, the authors did not comprehensively build the regulatory network, rather than adding some findings together.
- The quality of writing is poor, inconsistency in the naming of miRNAs and defining the abbreviations.
Change regulation network to regulatory network.
Line 20-22: I could not find the regulatory network in the manuscript, hope it does not figure 2. Did the authors perform some in-silico analyses?
Line 22: benefit early artificial selection and efficiency, it is not the clear the meaning of efficiency in the context.
Line 22-23: “provide new insight into the role of biomarkers and molecular genetics in chicken breeding.” It is not clear from the manuscript- what is the new insight from the manuscript. The authors might consider rewriting
Line 27: “is China's second most popular meat product”, the authors might remove it, it is not necessary to state in this context as the manuscript focuses on international readers.
Line 28-36: Please add the supporting references for this statement.
Line 32-33: it is not clear meaning “and the most active expression of genes involved in muscle development”; did the authors mean the most genes expressed in this stage or some other things?
Line 38-39: please provide supporting references.
Line 42-45: The authors should shortly summarise the classification of ncRNAs.
Line 45-47: I suggest the authors provide tables for summary numbers of ncRNAs (known and novel) that have been identified for the species.
I suggest the authors provide a short overview of the biology of skeletal development (a paragraph or even a section), and how they change in different stages. It will help the readers have a better view of the topic, before specifying it in the regulatory network.
Line 89: What is (SLDs)? Why did the authors use an abbreviation for it?
Line 90: Did the author mean all mutations or specific one “Once GHR gene mutations”. Please clarify
Line 94 and others: Change gene names to Italic
Line 98-99 and other: please do not use abbreviations if they appeared one or two times in the manuscript.
Line 113-115: Better including the supporting references.
Line 120: Why specify the “Chinese domestic chickens”
Line 124 and others: gga-miR- : Keep consistently when writing the miRNAs name; either having species names or not.
Line 138-140 and others: Please write the full names for development stages or define the abbreviation (E12, E16, E18, E20, D1) when using in the manuscript.
Table 1: Please write the full names of chicken breeds or define them
What did the authors mean: Organization, might change to tissues or organs?
Remove the column named year, it is not necessary to be included.
Figure 1: missing some – for the miRNA names?
The roles of ANKRD9 are not clear here, how it can promote the proliferation but inhibit differentiation.
Remove gga- in the gga-miR-3525 and others.
Where is the axis?
Line 229: What is “ce RNA “?
Line 242-243: “the specific functions of lncRNAs in skeletal muscle have received less attention”, I do not agree. It is because of the fact that functional validation of lncRNAs is harder than miRNAs.
Line 244: Could give a more accurate number “a number of lncRNAs “
Line 249-251: Please explore what is the finding in this research.
Line 254: they were unable to determine, or they just did not do it.
What did the author mean by “According to the latest research”?
Line 274-276: is this style suitable for the scientific paper “So far, we've found a lot of information about the levels of expression and functions of lncRNAs in the skeletal muscles of domestic chickens”
Also providing all the collected information in the supplementary files.
Line 438: Please remove it “, particularly among Chinese indigenous breeds”.
Line 444: It is not correct “Because the expression abundance of RNAs is relatively low and the measurement costs a lot, increasing the difficulty of studying their functional mechanisms” not all the ncRNAs are lowly expressed. The measure of them is not costly all time, it depends on the methods used.
Line 444-445: The authors should not generalize the information: miRNAs are conserved among species. Is the conservation related to the function among the species?
Line 447: Did the authors mean breeds?
Line 450: What did the authors mean by normal, might be healthy chicken is the better word.
There are some databases for the ncRNAs for chicken, the authors should search and update them in the manuscripts.
standardize ncRNA naming” please specify which types of ncRNAs, it is quite standard for miRNAs.
A lot of tools are available, which specific tools do the authors want to develop?
I do not understand why need to observe the point 4
Line 458-460: Why do we need to use these tools, the perspectives should be clearly stated
Round 2
Reviewer 1 Report
Revised manuscript is improved and authors have addressed the concerns.
There are couple of points-
- In the title, please remove "The". Start with "Regulation.......".
- In the Figure 1 and figure 2. Its a review article so readers would be benefited form understanding the models/schematics. I would recommend to elaborate the schematics. Both schematics have tons of molecules/proteins/components shown but legend barely mentions any of them. Please mention what to understand from those dense schematics. At the end of figure legend, consider putting the full form of the molecules/proteins/components used in figure.
Reviewer 2 Report
The authors have addressed my comments. The authors should also provide information about which miRNAs and lncRNAs should be focused on in future studies in the perspective section.
Line 543-557: The authors should also explore how this could be done for each suggestion instead of listing them. For instance, "upgrade specialized databases" which database and how to upgrade them, which types of information are needed to upgrade, etc.
For miRNAs and lncRNAs, the authors should not generalize information as miRNAs are better studied compared to lncRNAs.
The authors might look at some refs https://www.frontiersin.org/articles/10.3389/fvets.2020.578193/full#h8
https://www.ncbi.nlm.nih.gov/pmc/articles/PMC8002598/
https://pubmed.ncbi.nlm.nih.gov/27615279/
https://link.springer.com/article/10.1007/s00335-021-09928-7#Sec22
Some abbreviations are still not consistently used, for instance, the ncRNA abbreviation should use in line 14, not 15.
The quality of writing has not significantly improved. The authors did not proofread the manuscript well as some Non-English language still remains.
Please use the color highlighted instead of track-change for revision.
